# A novel approach of creating sustainable urban planning solutions that optimise the local air quality and environmental equity in Helsinki, Finland: The CouSCOUS study protocol

**Joanne C. Demmler**[1,2]*, **Ákos Gosztonyi**[1,2,3], **Yaxing Du**[4], **Matti Leinonen**[5], **Laura Ruotsalainen**[2,5], **Leena Järvi**[2,4], **Sanna Ala-Mantila**[1,2]

**1** Faculty of Biological and Environmental Sciences, University of Helsinki, Helsinki, Finland, **2** Helsinki Institute of Sustainability Science (HELSUS), University of Helsinki, Helsinki, Finland, **3** Helsinki Inequality Initiative (INEQ), University of Helsinki, Helsinki, Finland, **4** Institute for Atmospheric and Earth System Research (INAR), University of Helsinki, Helsinki, Finland, **5** Department of Computer Science, University of Helsinki, Helsinki, Finland

* joanne.demmler@helsinki.fi

**Funding:** LR, SA-M and LJ received funding by the Academy of Finland (https://www.aka.fi/en;

## Abstract

### Background

Air pollution is one of the major environmental challenges cities worldwide face today. Planning healthy environments for all future populations, whilst considering the ongoing demand for urbanisation and provisions needed to combat climate change, remains a difficult task.

### Objective

To combine artificial intelligence (AI), atmospheric and social sciences to provide urban planning solutions that optimise local air quality by applying novel methods and taking into consideration population structures and traffic flows.

### Methods

We will use high-resolution spatial data and linked electronic population cohort for Helsinki Metropolitan Area (Finland) to model (a) population dynamics and urban inequality related to air pollution; (b) detailed aerosol dynamics, aerosol and gas-phase chemistry together with detailed flow characteristics; (c) high-resolution traffic flow addressing dynamical changes at the city environment, such as accidents, construction work and unexpected congestion. Finally, we will fuse the information resulting from these models into an optimal city planning model balancing air quality, comfort, accessibility and travelling efficiency.

Academy of Finland grant numbers: 332177, 332179, 332178) and from the University of Helsinki (https://www.helsinki.fi/en) PROFI3 profiling grant. The funders had and will not have a role in study design, data collection and analysis, decision to publish, or preparation of the manuscript.

**Competing interests:** The authors have declared that no competing interests exist.

**Abbreviations:** ACHEM, 2D-Lagrangian Model for Aerosol Dynamics, Gas-Phase Chemistry and Radiative Transfer; AI, Artificial Intelligence; CARLA, Car Learning to Act Simulator; CouSCOUS, Sustainable urban development emerging from the merger of cutting-edge Climate, Social and Computer Sciences; DNN, Deep Neural Network; DRL, Deep Reinforcement Learning; ECWMF, European Centre for Medium-Range Weather Forecasts; ENFUSER, ENvironmental information FUsion SERvice; FIONA, Finnish Online Access (Remote Access System of Statistics Finland); FLEXPART, FLEXible PARTicle dispersion model; GAM, Generalized Additive Models; GIS, Geographic Information System; LES, Large Eddy Simulations; ML, Machine Learning; PALM, Parallelized Large-eddy Simulation Model; RL, Reinforcement Learning; SALSA, Sectional Aerosol module for Large Scale Applications.

## Introduction

Decreased air quality is one of the major environmental challenges cities worldwide face today and is estimated to cause 4.2 million premature deaths worldwide per annum [1]. Planning healthy environments for future populations, whilst considering the ongoing demand for urbanisation and provisions needed to combat climate change, remains a difficult task. This study combines the fields of artificial intelligence (AI), atmospheric and social sciences to provide urban planning solutions that optimise local air quality by applying novel methods and taking into consideration population structures and traffic flows.

Road traffic is a dominant factor in air pollution of both gaseous pollutants and particulate matter [2]. Unfortunately, the highest pollution levels are commonly seen at the pedestrian level [3] and might thus discourage healthy urban outdoors activities and active transport [4], which would otherwise alleviate traffic, noise and air pollution. The spatial attributes of the urban landscape (i.e. surface roughness of buildings) and turbulent mixing of the air can lead to inefficient pollutant transport from the street level under prevailing meteorological conditions [5]. The turbulent properties and pollutant transport of the flow are modified by these street canyons [6, 7] and their thermal properties [8, 9]. Although the pollutant sources and main features of pollutant dispersion in urban areas have largely been identified, we still poorly understand pollutant distribution at the micro-level of real urban surfaces and are lacking in air quality models that can efficiently solve their complex flow and pollutant distributions equations [10]. The 3D pollutant distributions within real urban neighbourhoods can be resolved using high-resolution air quality modelling, such as large eddy simulations [11–13].

Another important factor in the distribution of air pollutants is the traffic flow in a city. It is a key aspect of planning sustainable cities. When planning new traffic networks not only private and public transport need to be considered, but also cycle and pedestrian pathways as well as green spaces. Efficient traffic flow is essential in reducing traffic congestion, carbon and air pollutant emissions. Predicting traffic flow is, however, not a straightforward process as it is subject to large temporal fluctuations due to weather, roadworks or local events. Currently used machine learning (ML) methods that predict traffic flow are based on deep neural networks (DNNs) analysis [14]. These are based on semi-supervised methods in need of training data and therefore not feasible for the planning of new areas. Reinforcement learning (RL) is a novel ML method that uses a reward function and does not need a training dataset [15] and is thus better suited to planning of new city areas.

Exposure to air pollution might also have a larger impact on certain socio-economic groups which can indicate environmental inequality [16]. An unequal exposure distribution might worsen already existing health disparities between different population groups. Empirical research from the U.S. has established that individuals and communities with lower socioeconomic status are in many cases exposed to higher levels of pollutants [16, e.g. 17], although the opposite has been shown to be true for example in New York [18]. The built environment and access to blue-green spaces have an important impact on the mental health of urban dwellers [19–21]. Distance to facilities, street connectivity, safety and population density have all been shown to affect healthy walking behaviours (for example walking to schools [22, 23]), which in turn affect the use of public transport and driving behaviour and ultimately air pollution. Exposure might also differ by housing tenure type, although this might only be true for private housing [24]. Moreover, earlier research from the UK concluded that despite owning fewer cars, the poor drive older cars with higher emissions, and thus significantly contribute to air pollution [25]. However, more recent research [26] contradicts such conclusion and demonstrates an inverse relationship between transport-related emissions generation and poverty,

and further indicates (in line with [25]) that young children, young adults, and poor house-holds experience the highest levels of exposure. Further information on the socio-economic spatial inequality of pollutant exposure is needed as well as information on urban planning decisions aiming to alleviate or avoid such inequality.

Helsinki, the capital city of Finland, is experiencing a rapid population growth and a grow-ing demand in housing. The city has pledged the implementation of the Agenda 2030 Sustain-able Development Goals on local level [27] and aims to become carbon neutral by 2035. The city has a high percentage of active transport, with 77% of all journeys attributed to walking, cycling or public transport [28]. However, this still leaves hundreds of thousands private vehi-cle journeys at certain city areas every day. The city is currently in the early stages of designing and developing the so-called city boulevard areas around two major access roads. Additionally, we have limited and outdated knowledge about the existence of air pollution-related environ-mental inequalities in Helsinki. Research from 2000–2001 indicate lower levels of air pollution exposure with higher level of education and (gendered) occupational status [29, 30], however, the dynamic socio-spatial dimensions of environmental inequalities have never been studied in Finland to date.

According to Chi & Voss [31] small-area population forecasts are essential for sound local planning and decision making, but are dependent on a variety of factors, that are usually ignored in traditional forecasting methodologies. These relate particularly to neighbourhood characteristics and choices, such as accessibility to transportation and services [e.g. 32, 33], the physical environment [21, e.g. 34] and housing preferences [e.g. 35, 36]. Moreover, these pref-erences are likely to change with circumstances and age [e.g. 33, 37, 38]. It has also been shown that "desirable" neighbourhoods are more easy to predict than, for instance, rural areas [39]. The majority of these small-area population forecasts are, however, based on aggregated data at community or municipality level and not on micro-level data of households or neighbour-hoods and are thus likely to misinterpret the neighbourhood context.

In this study we are going to combine high-resolution traffic flow and pollutant distribution models with micro-level population data to help design a new "city boulevards" neighbour-hood in Helsinki in such a way as to minimise local air pollutant concentrations, accessibility, travelling efficiency and environmental inequality.

## Methods and analysis

### Study design, aims and hypotheses

This study combines three different paths: (i) we are going to use high-resolution micro-level urban, demographic and meteorological data to create a retrospectively linked electronic cohort for all residents of Helsinki (60˚ 12' 49.395" N, 24˚ 53' 3.8502" E) between 2010 and 2020 (depending on individual dataset coverage); (ii) an air quality model will be built, based on detailed Vihdintie boulevard layouts and high-resolution 3D surfaces (see Fig 1); (iii) a deep learning-based traffic flow model will be developed using historical traffic data from an area with similar characteristics as the Vihdintie boulevard (iv) fusing models (i)-(iii), a sophis-ticated tool, based on Reinforcement Learning, will be built, that outputs advice about how the traffic should be arranged for the Vihdintie boulevard study area for optimizing comfort, min-imal pollution emissions, accessibility and travelling efficiency; this simulation will be based on a realistic representation of the planned city area layout.

Our study has four main aims:

1. Quantify the possible co-locations of urban population groups, urban amenities, and air quality effects both in the present and future situation.

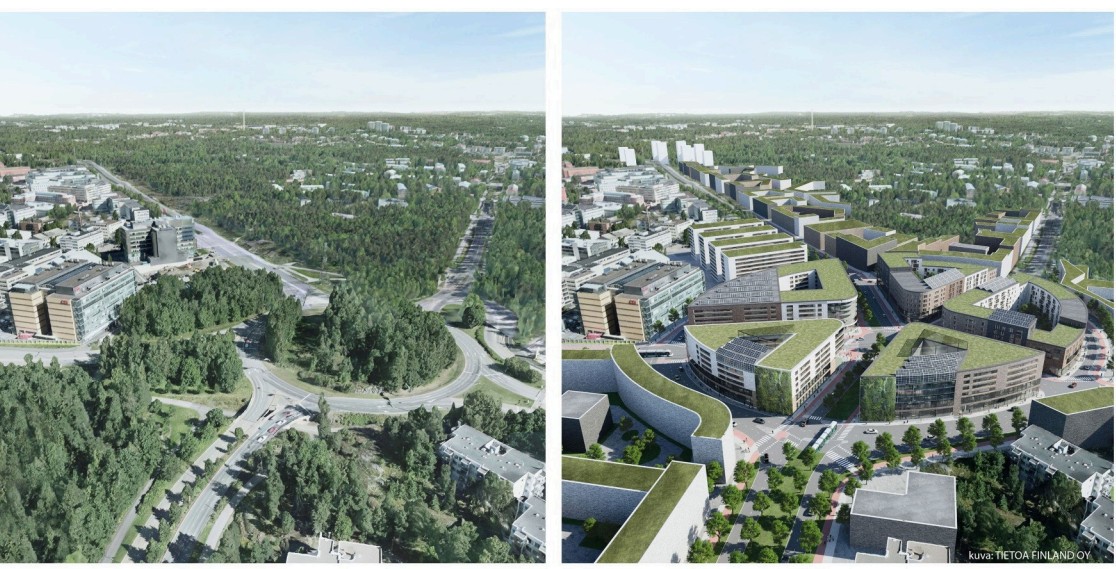

**Fig 1.** Current (left) and future envisioned view (right) of the Vihdintie Boulevard. Photo: Tietoa Finland Oy (Accessed on 23/06/2121 from https://kerrokantasi.hel.fi/bulevardikaupunkia/gMuQSTFoRuyx8CixB5hYMK171lbbaxEW?lang=en).

2. Understand the high-resolution spatial and temporal variability of air quality at street space in different planning and traffic flow scenarios using a novel LES based air quality model that can account in detail urban neighbourhoods.

3. Create a tool based on artificial intelligence methods providing suggestions for traffic planning for optimizing comfort, air quality, accessibility and travelling efficiency.

4. Estimate and provide recommendations for the most optimal planning solution from air quality, comfort, accessibility and travelling efficiency aspects.

The project will attempt to answer the following main research questions:

- What are the main factors in the urban structure (both thermal and mechanical effects) causing formation of air quality hotspots?

- Is there a relationship between socioeconomic status and air pollution exposure? What explains the possible relationship?

- How can we predict future population using AI-modelling utilizing detailed micro-level data on population and various sources of GIS (Geographic Information System) data about built environment structures?

- How should the reward function in Reinforcement Learning be composed to reliably accommodate all information of very different nature (traffic, population, air quality) and to provide finally a good suggestion for city planning?

- What is the most optimal urban planning choice including traffic scenario for creating the best air quality in the planned city boulevard in Helsinki?

How well can the Machine Learning models built, trained and tested in one city be scaled for other cities in Europe?

## Modelling platforms and datasets

**A) Micro-level data modelling.** We will be using the Finnish Online Access (FIONA) Remote Access System of Statistics Finland to access the Statistics Finland micro-level population data relating to demographics and housing. These data will be augmented with data from other sources, such as housing data and air pollution data. The full list of the data and their sources is presented in Table 1.

**B) High-resolution air quality modelling.** The highly resolved air quality distributions in the planned city boulevard in Helsinki will be simulated and analysed using the LES model PALM, which presents the state-of-the-art in urban atmospheric modelling [43]. It is the first model of its kind that allows to simulate both detailed aerosol dynamics, aerosol and gas-phase chemistry together with detailed flow characteristics. In PALM, aerosol dynamics are simulated using the Sectional Aerosol module for Large Scale Applications (SALSA) [44] allowing to examine in detail deposition, nucleation coagulation and condensation of aerosol particles, and basic gas compounds. PALM has successfully been used to examine high-resolution pollutant concentrations in e.g. Helsinki and Berlin [12, 47] and particularly how different urban planning solutions can lower local pollutant concentrations [11, 48].

**C) High-resolution traffic flow modelling.** The traffic flow model will be developed using deep learning methods. The model will generate traffic addressing different dynamic changes in the city environment, arising from changes in weather and seasons, accidents,

**Table 1. Data and data sources for the CouSCOUS project.**

| Data type | Source | File format | Size |
|---|---|---|---|
| Traffic data, vehicle numbers per vehicle type | City of Helsinki (https://hri.fi/en_gb/) | csv | ~100MB |
| Air quality, meteorological and turbulence data | City of Helsinki (https://hri.fi/en_gb/) & European collaborator cities | csv / txt | ~50MB |
| ENFUSER 2.0 air quality data | Finnish Meteorological Institute (https://en.ilmatieteenlaitos.fi/) Johansson et al. [40, 41] | netcdf | ~ 265GB |
| Climate data | European branch of the Coordinated Regional Climate Downscaling Experiment (https://www.euro-cordex.net/) | netcdf | >1 GB |
| 3D surface model | Nordbo et al. [42] | raster | ~10MB |
| Emission factors | City of Helsinki (https://hri.fi/en_gb/) & national Lipasto database (http://lipasto.vtt.fi) | csv | ~10MB |
| ECWMF re-analysis and FLEXPART datasets for ADCHEM runs | ECMWF (https://www.ecmwf.int/) & FLEXPART (https://www.flexpart.eu/) websites | netcdf | > 1GB |
| Population statistical data & Open data for the Helsinki Region | Statistics Finland micro-level data (remote access FIONA) (https://www.stat.fi/tup/mikroaineistot) & Statistics Finland gridded data (https://www.tilastokeskus.fi/tup/ruututietokanta/index_en.html) & Helsinki Region Infosphere (https://hri.fi/en_gb/) & Avoin Data (https://www.hsy.fi/avoindata) & Helsinki Region Service Map (https://palvelukartta.hel.fi/en/) | txt, .xlsx, csv, geojson, .shp | |
| Population & built environment statistical data | LIITERI-database by Finnish Environmental Institute SYKE (https://www.syke.fi/en-US) | txt, .xlsx, csv, geojson, .shp | |
| Planning data from City of Helsinki | City of Helsinki Planning Department | | |
| Input and output data of the PALM model and ADCHEM model runs | PALM: Maronga et al. [43], Kurppa et al. [44] ADCHEM: Roldin et al. [45] Data produced | netcdf files, model scripts in fortran, scripts using common programming languages such as Python | ~10MB |
| Analysis of traffic flow with different parameters based on the DRL runs presented as optimal ratios of each traffic mode | Data produced | pdf | ~5 MB |
| Examples of the CARLA analysis runs | Dosovitskiy et al. [46] https://www.unrealengine.com/en-US/ Data produced | MP4 | ~100 MB |

construction work and other events changing the traffic flow. Traffic flow modelling is a challenging task due to its spatio-temporal nature and because it is prone to influence by the above-mentioned external factors and many more. The state-of-the-art methods do not solve the challenges at sufficient level, and therefore there is a need to develop sophisticated deep learning methods for traffic flow prediction [49]. The generated traffic will be fed to the CARLA open source, flexible urban driving simulator [46], which was originally designed for aiding the research on automated vehicles. The simulator will be augmented with the 3D city models, obtained from the plans for the Vihdintie boulevard.

**D) Fusion of the models into a city planning tool.** At present, the use of Reinforcement Learning (RL) in complex tasks is quite limited due to the complexity of its implementation. Because the goal of our project is not to do traditional traffic flow prediction using existing traffic data, but to combine traffic data with data of very different nature, air quality and population-related, research for developing sophisticated RL methods for city planning is needed [50]. Model Based Reinforcement Learning is anticipated to be the best ML method to be used in traffic control due to its somehow predictive nature [51]. Therefore, we will develop novel RL methods for combining traffic data and novel LES modelling to optimise local air quality and to fuse the population data with them.

## Data management plans

Each consortium party will use their file services to store data gathered during the project. During the project the data created for training and resulting from the deep neural networks will be stored in a MongoDB database that will be set up into CSC's ePouta cloud service (https://research.csc.fi/). CSC will back up the data.

The model calculations will mainly be made on the CSC IT Center supercomputers and input and output model data will be uploaded to the secure archive storage of CSC from the "work directories" where the actual calculations are run. Some of these data will also be downloaded to local computers if the datasets are small enough. CSC is responsible for data backup. Individual-level microdata from Statistics Finland will be handled and stored according to their rules and using their distance-use connection. All results will be stored on protected servers and nothing that is identifiable will be published.

## Status and timeline of the study

We are currently awaiting access to the micro-level data. We have conducted several test runs of high-resolution air quality modelling for Malmö, Sweden using PALM and are now preparing micro-level input files (down to 1 m resolution) for PALM simulations to study the influences of different urban planning solutions on street-level air quality. We will then prepare the input files for the boulevards of interest in Helsinki and conduct dozens of PALM runs under different scenarios. In this way, we can provide enough data for the Reinforcement Learning to optimise local air quality. For traffic modelling and simulation, the data has been chosen and the work to combine the different sources of data has been started. Development of the traffic generation system is ongoing, the network and training decisions have been made. The anticipated timeline of the study is shown in Table 2.

## Data analysis

**A) Micro-level data modelling.** We will study the patterns of locations and relocations of population, and potential relationships they have with air quality at the city scale. The focus is on possible co-locations of certain socioeconomic groups, their propensity to move, and local emissions and air quality. This analysis leans on the tradition of environmental equity [52], yet

**Table 2. Proposed timeline of the project.**

| Task | 2020 | 2021 | | 2022 | | 2023 | | 2024 |
|---|---|---|---|---|---|---|---|---|
| | H2 | H1 | H2 | H1 | H2 | H1 | H2 | H1 |
| **Data collection and dialogue with stakeholders** | ▓ | ▓ | ▓ | ▓ | ▓ | | | |
| **Recruitment** | ▓ | ▓ | | | | | | |
| **Environmental justice and population forecasting** | | | | | | | | |
| 1) Review of literature | | ▓ | ▓ | ░ | ░ | | | |
| 2) Access to micro-level data | | ░ | ▓ | ▓ | ▓ | | | |
| 3) Modelling | | | | ░ | ▓ | | | |
| **LES model** | | | | | | | | |
| 1) Modelling | | | | | ▓ | ▓ | ░ | |
| 2) Data analysis | | | | | ░ | ▓ | ▓ | |
| **DRL/DNN modelling** | | | | | | | | |
| 1) Traffic generation | | ▓ | ▓ | ▓ | | | | |
| 2) Deep reinforcement learning | | | | | | ▓ | ▓ | ▓ |
| **Synthesis and scalability** | | | | | ▓ | ▓ | ▓ | ▓ |

the approach is improved by using micro-level population data and modelled environmental exposure at very granular resolution, in contrast to traditional approach of self-reported survey-data or proxies, such as distances to roads. The key data sources for improvements are high-quality FMI-ENFUSER air quality data [40, 41], and micro-level population data from Statistics Finland. Descriptive and visual spatial analyses of the population and environmental quality are done. Attention is also paid to the types of neighbourhoods, their amenities and built environment structures, to see whether e.g. newly built residential areas differ from the typical suburbs built in the 60s and 70s. Similarly, the locations of certain services, such as schools and kindergartens, is combined with the air-quality data, to see whether there are differences in exposure.

We then will run and compare more advanced models to explain air quality exposure and different inequality metrics. More traditional spatial models, such as multivariate models and multilevel models, are tested along with spatial modelling options including, e.g. spatial generalized additive models (GAM) and spatial autoregressive models. The key issue is to take into account the spatial structure of air pollution data (spatial autocorrelation), a trait that is often neglected in previous studies [see 53].

A model for urban population forecasting is developed using micro-level data on population and information on various built environment structures, such as accessibility data. The model is tested using the existing data on population development and then used to predict the population structure in the investigated city neighbourhood given the plans. Population structure prediction also includes prediction of car-ownership rates, which will be used in creating socio-economic mobility types. The applicability of machine learning techniques to predict the population is investigated, and the results are compared to more traditional statistical forecasting methods. Using and understanding the novel methods and their usefulness for population forecasting is beneficial not only for understanding the health effects, but also for service planning purposes. In the realm of population forecasts, ML techniques have been applied already a couple of times: to forecast the prevalence non-communicable diseases at US state level [54] as well as residential relocation patterns in Seoul [55]. The different single-parameter models are developed to predict the age distribution in the planned neighbourhood on building resolution as step one, also other socioeconomic variables can be estimated with

similar techniques. In step two, we work towards a multi-parameter model, which would give more detailed individual-level estimations on simulated residents.

**B) High-resolution air quality modelling.**  One of the major tasks in LES based air quality modelling is the creation of modelling domain and selection of appropriate boundary conditions. In the model runs, one-way nesting capability of PALM will be used. A higher resolution child domain (e.g. 1536x1536x96 m3 with 1 m resolution [12]) will cover the neighbourhood of interest and within this domain the aerosol processes will only be treated. The parent domain (e.g. 2304x2304x288 m$^3$ with 2–4 m resolution) and root domain (6912x6912x606 m$^3$ with 8–12 m resolution) will cover larger areas to allow the flow to adjust to the urban surface. Dynamic meteorological boundary conditions will be used in the root domain. Within the child domain, detailed city boulevard layouts will be obtained from the City of Helsinki, whereas for the parent domain 3D surface model based on Lidar data will be used [56]. Climate scenarios and present-day knowledge on background concentrations will be used as model boundary conditions. Other boundary conditions include information about vehicle fleet and emission factors which can be obtained from a similar approach as used by [11].

The ongoing transition from gasoline vehicles to electric vehicles will be taken into account in the calculation of emission factors based on the predictions provided by traffic authorities [11]. There will be significant amount of time when both vehicle types will co-exist during which particle emissions still take place. And even if all vehicles would be electric vehicles, in northern latitudes resuspended road dust causes significant source for particles in springtime. As electric vehicles are heavier than gasoline cars due to batterie) the road dust emissions are expected to increase in future. This will be included in the calculation of emission factors given to the model [11].

The spatial and temporal variability of aerosol particles (size distribution, mass) and gaseous compounds (NO, NO2 and O3) will we simulated for selected representative days in summer and winter periods in the planned city boulevard. Model runs will be made for different urban planning alternatives with varying traffic scenarios with the aim of creating reward functions for the deep reinforcement learning (DRL) algorithm. The exact number of the modelled alternatives depends on plausible options the city of Helsinki is considering but will be around 20–40 model runs. From these runs, the spatial and temporal variability of the concentrations fields and hotspots will be evaluated and controlling factors determined. Due to the highly variable pollutant fields and great amount of data the model is providing, together with the stakeholders we need to carefully design a ranking system to decide for which areas (i.e. pavements, tram stops, different building floors) the reward functions will be created.

**C) High-resolution traffic flow modelling.**  At present, CARLA enables simulation of the traffic flow at a microscopic level, namely simulates the movement of individual vehicles and generates other actors in the traffic using standard Unreal Engine's (https://www.unrealengine.com/en-US/what-is-unreal-engine-4) vehicle model and their motion using a basic controller defining their behaviour. To achieve our research goals, we need to have more intelligence on the generation of vehicles, pedestrians and bicycles into traffic than just the existing random process.

A DNNs method will be developed for generating the prediction of the amount of people in the traffic. For this, a number of socio-economic mobility profiles will be constructed based on micro-level socio-economic data and car ownership data, which will feed into training the learning algorithm, together with historical traffic data. The result will be a DNNs algorithm that has learned to predict the number of travellers of all transport profiles (vehicles, public transportation, pedestrian, bicyclist) passing the city area conditional on the weather, time of day, season, events, construction works and demographic structure at the area. Distributional

effects of consequent predicted changes in air pollution will be examined to aid decision making on identifying optimal urban planning scenarios.

**D) Fusion of the models into a city planning tool.**   The outcome of our research will be a DRL-based algorithm fusing the information obtained from the models developed in steps a)-c). The most challenging part of the DRL algorithm development is the design of the reward function [15]. Some research using RL for traffic flow prediction has been done, but their approach is very simplistic, e.g. they use very simple reward functions [50]. Our solution will not only predict the traffic flow, but will give recommendations of how the traffic should be organized (spatially and considering different traffic modes) for optimizing the sustainability aspects of the area. Therefore, the reward function will be formed as a combination of smallest pollution effects, efficient commuting, liveability, accessibility, and other factors agreed with all stakeholders.

The interplay of all three analysis paths is shown in Fig 2.

## Synthesis and scalability

The final aim of the project is to provide recommendations for the planned city boulevard in Helsinki and examine the scalability of the developed methods to other neighbourhoods in different cities. The short-term air pollutant concentrations with different planning alternatives in the planned city boulevard accounting for accessibility, travelling efficiency and socio-economic structures are estimated and recommendations for urban planning provided. For example, with one urban built environment structure certain population groups are more likely to move in, creating particular traffic scenarios furthermore impacting the local air quality and

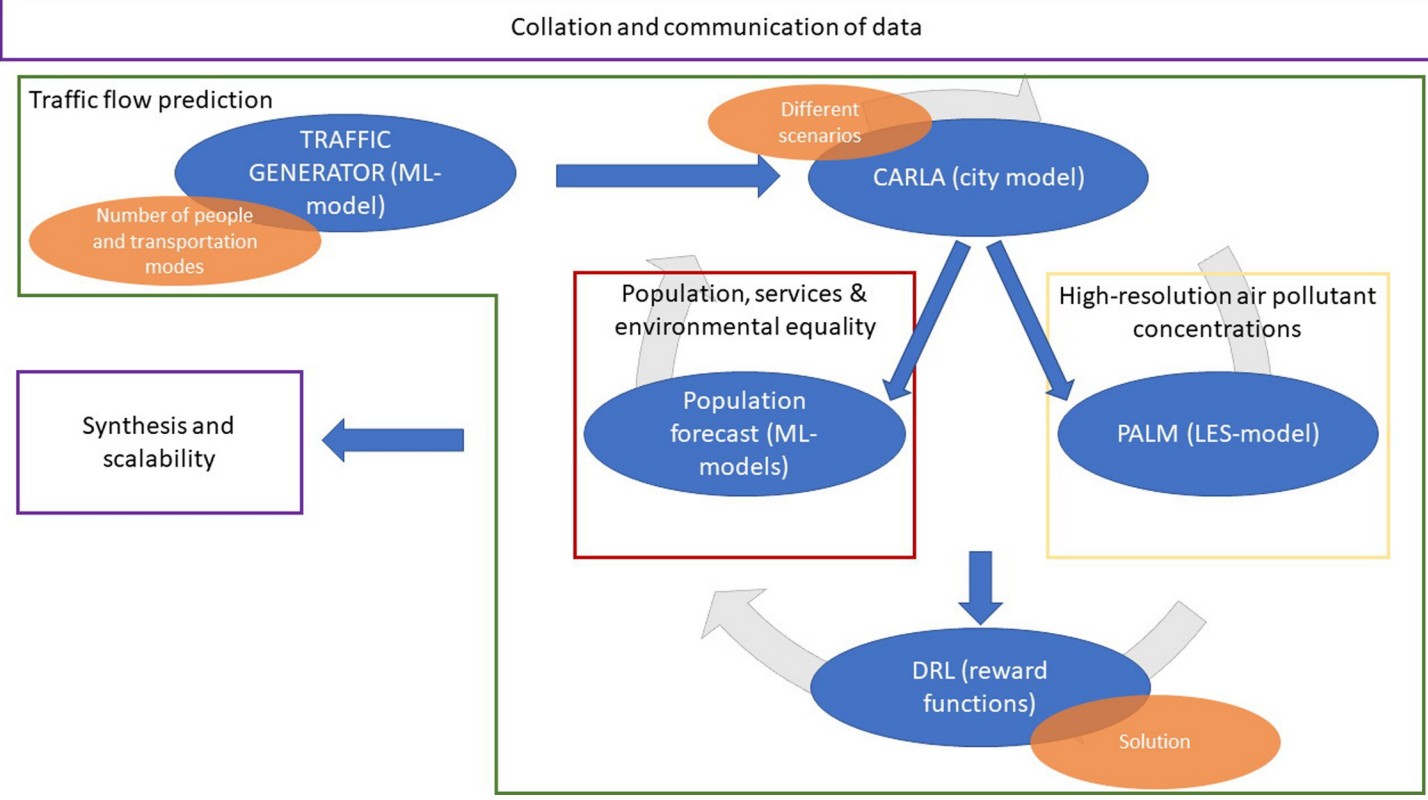

**Fig 2. Modelling interconnection between the different research teams and methods.**

air pollution exposure to the different population groups. The local air quality distributions are largely dependent on the built environment structure (including buildings, trees, etc.) and therefore we will also examine the scalability of the DRL algorithms with different neighbourhoods. Some information is provided with the different urban planning alternatives of the planned city boulevard, but we will furthermore examine the performance of the algorithms with existing street canyons in Helsinki. The neighbourhood was selected in addition to PALM produced air quality fields.

In addition to Helsinki, the scalability of the DRL algorithm in other cities will be studied. The high-resolution distributions of air pollutant concentrations in neighbourhoods in Malmö (Sweden), Turin (Italy) and Guildford (UK) will be simulated for selected few hour periods using PALM. The model outputs together with known traffic distributions and socio-economic information provided via collaboration with the respective researchers allow us to examine how well the algorithms developed in Helsinki will reproduce the air quality fields in these cities with different neighbourhoods. The cities were selected as there are also measured data available allowing to evaluate the model performance at the same time when testing the algorithms. The ADCHEM model [45] will be used to provide boundary conditions for pollutants whereas other needed data with 3D surface maps, emissions factors and meteorology will be obtained from the respective researchers.

## Ethics and dissemination

No ethical or other conflicts are expected in the experiments or material, and no experiments are performed that would require permissions. Work with databases, data, codes, and models will not cause any intellectual property infringement, as none of these are subject to official restrictions.

Sensitive population data will be analysed within the FIONA remote desktop of Statistics Finland. Output of these data are subject to disclosure control. Results for this project will be disseminated to city developers and wider audiences.

## Discussion

In this study we will combine high-resolution traffic flow and pollutant distribution models with micro-level population data to help design a new "city boulevards" neighbourhood in Helsinki in such a way as to minimise local air pollutant concentrations, accessibility, travelling efficiency and environmental inequality. However, combining these separate fields of science is a high-risk, yet potential high-reward, strategy in itself, and the ability of the project to provide usable tools for practice and local decision-making needs constant and careful attention during the execution of the study.

### Limitations of the study

Factors that might affect air pollution exposure might not be available to us. For instance, we have information for buildings, but not for individual households or floors within buildings [57]. We also have no information on the type of residential heating [58] or on indoor air quality. Only outdoor air pollution is included in our study. Furthermore, it is still to some extent unclear how well scalability of the models to other neighbourhoods and cities work, as the data availability we have for Helsinki can differ from that in other cities.

### Project dissemination

This project will be conducted in close collaboration with policy makers and city planners and developers of the City of Helsinki. Aside from presenting future findings to the academic

community we will also disseminate them to the wider public via social media and a project website and blog (https://www2.helsinki.fi/en/projects/couscous).

## Acknowledgments

We would like to thank the City of Helsinki for the opportunity to work together on this project.

## Author Contributions

**Conceptualization:** Laura Ruotsalainen, Leena Järvi, Sanna Ala-Mantila.

**Formal analysis:** Joanne C. Demmler, Ákos Gosztonyi, Yaxing Du, Matti Leinonen, Laura Ruotsalainen, Leena Järvi, Sanna Ala-Mantila.

**Funding acquisition:** Laura Ruotsalainen, Leena Järvi, Sanna Ala-Mantila.

**Investigation:** Joanne C. Demmler, Ákos Gosztonyi, Yaxing Du, Matti Leinonen, Laura Ruotsalainen, Leena Järvi, Sanna Ala-Mantila.

**Project administration:** Laura Ruotsalainen, Leena Järvi, Sanna Ala-Mantila.

**Supervision:** Laura Ruotsalainen, Leena Järvi, Sanna Ala-Mantila.

**Writing – original draft:** Joanne C. Demmler.

**Writing – review & editing:** Joanne C. Demmler, Ákos Gosztonyi, Yaxing Du, Matti Leinonen, Laura Ruotsalainen, Leena Järvi, Sanna Ala-Mantila.

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
