## [Decision Letter · Decision Letter 0]

31 Aug 2021

PONE-D-21-21473

A novel approach of creating sustainable urban planning solutions that optimise the local air quality and environmental equity in Helsinki, Finland: the CouSCOUS study protocol

PLOS ONE

Dear Dr. Demmler,

Thank you for submitting your manuscript to PLOS ONE. After careful consideration, we feel that it has merit but does not fully meet PLOS ONE’s publication criteria as it currently stands. Therefore, we invite you to submit a revised version of the manuscript that addresses the points raised during the review process.

We received very positive feedbacks on your manuscript by both reviewers but there are only minor issues to be considered in the revisions as highlighted by Reviewer 2. We expect you to do these revisions and re-submit your manuscript shortly.

We look forward to receiving your revised manuscript.

Kind regards,

Eda Ustaoglu, PhD

Academic Editor

PLOS ONE

**a)** If there are ethical or legal restrictions on sharing a de-identified data set, please explain them in detail (e.g., data contain potentially sensitive information, data are owned by a third-party organization, etc.) and who has imposed them (e.g., an ethics committee). Please also provide contact information for a data access committee, ethics committee, or other institutional body to which data requests may be sent.

We will update your Data Availability statement on your behalf to reflect the information you provide."

4. Thank you for stating the following in the Financial Disclosure of your manuscript:

“LR, SA-M and LJ received funding by the Academy of Finland (https://www.aka.fi/en; Academy of Finland grant numbers: 332177, 332179, 332178) and from the University of Helsinki (https://www.helsinki.fi/en) PROFI3 profiling grant. The funders had and will not have a role in study design, data collection and analysis, decision to publish, or preparation of the manuscript."

We note that you have provided funding information that is not currently declared in your Funding Statement. However, funding information should not appear in the other areas of your manuscript. We will only publish funding information present in the Funding Statement section of the online submission form.

“LR, SA-M and LJ received funding by the Academy of Finland (https://www.aka.fi/en; Academy of Finland grant numbers: 332177, 332179, 332178) and from the University of Helsinki (https://www.helsinki.fi/en) PROFI3 profiling grant. The funders had and will not have a role in study design, data collection and analysis, decision to publish, or preparation of the manuscript”

Reviewers' comments:

Reviewer's Responses to Questions

**Comments to the Author**

1. Does the manuscript provide a valid rationale for the proposed study, with clearly identified and justified research questions?

Reviewer #1: Yes

Reviewer #2: Yes

2. Is the protocol technically sound and planned in a manner that will lead to a meaningful outcome and allow testing the stated hypotheses?

Reviewer #1: Yes

Reviewer #2: Yes

3. Is the methodology feasible and described in sufficient detail to allow the work to be replicable?

Reviewer #1: Yes

Reviewer #2: Yes

4. Have the authors described where all data underlying the findings will be made available when the study is complete?

Reviewer #1: Yes

Reviewer #2: Yes

5. Is the manuscript presented in an intelligible fashion and written in standard English?

Reviewer #1: Yes

Reviewer #2: Yes

6. Review Comments to the Author

You may also provide optional suggestions and comments to authors that they might find helpful in planning their study.

Reviewer #1: I have greatly enjoyed reading this study protocol on such an interesting multidisciplinary topic. I cannot wait to read the results and see the AI tool in action. My only recommendation is to add urban design experts to your research team to help with the urban design details and variables that could be included in the model to improve the results.

Reviewer #2: The paper is the protocol for a substantial and sophisticated study of air pollution and urban planning in one city. The paper is very well written, thoroughly grounded in existing literature, and detailed in explaining the aims, proposed data sources and methods for the study. I strongly support publication and have only two very minor comments.

I was unclear how the study would deal with the increasing move to electric vehicles which have very different pollution profiles compared with those reliant on fossil fuels. There is a comment about vehicle fleet and emissions factors (p22, line334) but I didn’t see an explicit recognition of this. It would be helpful to have a sentence on how the study plans to cope with the uncertainties around rates of transition here.

I was unclear how or whether the study will capture distributional effects or differences by socio-economic status. I think this is primarily through looking at where people live rather than how and when they move through urban spaces. There is a comment (p23, line356) that the study has data on socio-economic mobility profiles but the same paragraph states only that models will predict numbers of travellers on different modes, not broken down by socio-economic status. A sentence or two elaborating further how distributive impacts will be considered would be welcome.

7. PLOS authors have the option to publish the peer review history of their article (what does this mean?). If published, this will include your full peer review and any attached files.

Reviewer #1: **Yes: **Dr. Samineh Ansari

Reviewer #2: No

---

## [Author Response · Author response to Decision Letter 0]

15 Oct 2021

Reviewer #1: I have greatly enjoyed reading this study protocol on such an interesting multidisciplinary topic. I cannot wait to read the results and see the AI tool in action. My only recommendation is to add urban design experts to your research team to help with the urban design details and variables that could be included in the model to improve the results. 

Thank you for the positive feedback. We are working closely with the City of Helsinki, who are advising us on urban design and variables of interest.

Reviewer #2: The paper is the protocol for a substantial and sophisticated study of air pollution and urban planning in one city. The paper is very well written, thoroughly grounded in existing literature, and detailed in explaining the aims, proposed data sources and methods for the study. I strongly support publication and have only two very minor comments.

Thank you for the positive feedback.

Reviewer #2: I was unclear how the study would deal with the increasing move to electric vehicles which have very different pollution profiles compared with those reliant on fossil fuels. There is a comment about vehicle fleet and emissions factors (p22, line334) but I didn’t see an explicit recognition of this. It would be helpful to have a sentence on how the study plans to cope with the uncertainties around rates of transition here.

The reviewer is right that naturally the amount of electric vehicles will increase in future which will reduce traffic related pollutant emissions. However, there will likely be long transition period when both gasoline and electric vehicles will be present in the vehicle fleet. This reduction in can be taken into account in the emission factors like was made for example in our earlier study by Karttunen et al. (2020). In Northern latitudes road dust resuspended by traffic causes significant source for aerosols in spring time and as electric vehicles are heavier than gasoline cars (due to batteries) the road dust emissions are expected to increase in future. Also these can be taken into account in the emission factor inputs to the model (see Karttunen et al. 2020)

We have added the following sentence to the manuscript (paragraph starting at line 421):

“The ongoing transition from gasoline vehicles to electric vehicles will be taken into account in the calculation of emission factors based on the predictions provided by traffic authorities (Karttunen et al. 2020). There will be significant amount of time when both vehicle types will co-exist during which particle emissions still take place. And even if all vehicles would be electric vehicles, in northern latitudes resuspended road dust causes significant source for particles in springtime. As electric vehicles are heavier than gasoline cars due to batterie) the road dust emissions are expected to increase in future. This will be included in the calculation of emission factors given to the model (Karttunen et al. 2020).”

Reviewer #2: I was unclear how or whether the study will capture distributional effects or differences by socio-economic status. I think this is primarily through looking at where people live rather than how and when they move through urban spaces. There is a comment (p23, line356) that the study has data on socio-economic mobility profiles but the same paragraph states only that models will predict numbers of travellers on different modes, not broken down by socio-economic status. A sentence or two elaborating further how distributive impacts will be considered would be welcome. 

Thank you for this comment. We do not have any direct data on how people move, but we can model the changes over time, e.g. changes in socio-economic status, changes in neighbourhood composition or possibly household composition (only possible for densely populated areas). Different levels of movement will also be included in the traffic model.

We will clarify this in the text as follows (paragraph starting at line number 447):

“A DNNs method will be developed for generating the prediction of the amount of people in the traffic. For this, a number of socio-economic mobility profiles will be constructed based on micro-level socio-economic data and car ownership data, which will feed into training the learning algorithm, together with historical traffic data. The result will be a DNNs algorithm that has learned to predict the number of travellers of all transport profiles (vehicles, public transportation, pedestrian, bicyclist) passing the city area conditional on the weather, time of day, season, events, construction works and demographic structure at the area. Distributional effects of consequent predicted changes in air pollution will be examined to aid decision making on identifying optimal urban planning scenarios.”

---

## [Decision Letter · Decision Letter 1]

2 Nov 2021

A novel approach of creating sustainable urban planning solutions that optimise the local air quality and environmental equity in Helsinki, Finland: the CouSCOUS study protocol

PONE-D-21-21473R1

Dear Dr. Demmler,

We’re pleased to inform you that your manuscript has been judged scientifically suitable for publication and will be formally accepted for publication once it meets all outstanding technical requirements.

Kind regards,

Eda Ustaoglu, PhD

Academic Editor

PLOS ONE

Additional Editor Comments (optional):

Reviewers' comments:

Reviewer's Responses to Questions

**Comments to the Author**

1. Does the manuscript provide a valid rationale for the proposed study, with clearly identified and justified research questions?

Reviewer #1: Yes

Reviewer #2: Yes

2. Is the protocol technically sound and planned in a manner that will lead to a meaningful outcome and allow testing the stated hypotheses?

Reviewer #1: Yes

Reviewer #2: Yes

3. Is the methodology feasible and described in sufficient detail to allow the work to be replicable?

Reviewer #1: Yes

Reviewer #2: Yes

4. Have the authors described where all data underlying the findings will be made available when the study is complete?

Reviewer #1: Yes

Reviewer #2: Yes

5. Is the manuscript presented in an intelligible fashion and written in standard English?

Reviewer #1: Yes

Reviewer #2: Yes

6. Review Comments to the Author

You may also provide optional suggestions and comments to authors that they might find helpful in planning their study.

Reviewer #1: I appreciate the authors' clarifications, especially concerning the integration of socio-economic mobility profiles in the algorithm, and I strongly support publication of this paper.

Reviewer #2: Thanks to the authors for addressing my minor comments. I remain strongly supportive of publication.

7. PLOS authors have the option to publish the peer review history of their article (what does this mean?). If published, this will include your full peer review and any attached files.

Reviewer #1: No

Reviewer #2: No

---

## [Editor Report · Acceptance letter]

24 Nov 2021

PONE-D-21-21473R1 

A novel approach of creating sustainable urban planning solutions that optimise the local air quality and environmental equity in Helsinki, Finland: the CouSCOUS study protocol 

Dear Dr. Demmler:

I'm pleased to inform you that your manuscript has been deemed suitable for publication in PLOS ONE. Congratulations! Your manuscript is now with our production department. 

Kind regards, 

on behalf of

Dr. Eda Ustaoglu 

Academic Editor

PLOS ONE